# Simultaneous Measurement of Urinary Trimethylamine (TMA) and Trimethylamine *N*-Oxide (TMAO) by Liquid Chromatography–Mass Spectrometry

**DOI:** 10.3390/molecules25081862

**Published:** 2020-04-17

**Authors:** Xun Jia, Lucas J. Osborn, Zeneng Wang

**Affiliations:** 1Department of Cardiovascular & Metabolic Sciences, Lerner Research Institute, Cleveland Clinic, 9500 Euclid Ave, Cleveland, OH 44195, USA; jiax2@ccf.org (X.J.); osbornl@ccf.org (L.J.O.); 2Department of Molecular Medicine, Cleveland Clinic Lerner College of Medicine, Case Western Reserve University, Cleveland, OH 44106, USA

**Keywords:** trimethylamine (TMA), trimethylamine *N*-oxide (TMAO), biomarker, urine, liquid chromatography–mass spectrometry (LC/MS)

## Abstract

Trimethylamine (TMA) is a gut microbial metabolite—rendered by the enzymatic cleavage of nutrients containing a TMA moiety in their chemical structure. TMA can be oxidized as trimethylamine *N*-oxide (TMAO) catalyzed by hepatic flavin monooxygenases. Circulating TMAO has been demonstrated to portend a pro-inflammatory state, contributing to chronic diseases such as cardiovascular disease and chronic kidney disease. Consequently, TMAO serves as an excellent candidate biomarker for a variety of chronic inflammatory disorders. The highly positive correlation between plasma TMAO and urine TMAO suggests that urine TMAO has the potential to serve as a less invasive biomarker for chronic disease compared to plasma TMAO. In this study, we validated a method to simultaneously measure urine TMA and TMAO concentrations by liquid chromatography–mass spectrometry (LC/MS). Urine TMA and TMAO can be extracted by hexane/butanol under alkaline pH and transferred to the aqueous phase following acidification for LC/MS quantitation. Importantly, during sample processing, none of the nutrients with a chemical structure containing a TMA moiety were spontaneously cleaved to yield TMA. Moreover, we demonstrated that the acidification of urine prevents an increase of TMA after prolonged storage as was observed in non-acidified urine. Finally, here we demonstrated that TMAO can spontaneously degrade to TMA at a very slow rate.

## 1. Introduction

Trimethylamine *N*-oxide (TMAO), the oxidative product of trimethylamine (TMA), which is dependent on gut microbiota, has gained wide interest due to its pro-atherogenic and pro-thrombotic properties [1,2,3,4]. Circulating levels of TMAO can predict future risk for major adverse cardiac events, myocardial infarction, stroke or death [5]. Apart from the clinical relevance to cardiovascular disease, TMAO has also been reported to contribute to chronic kidney disease progression and has been linked to type II diabetes mellitus, a key feature of the human metabolic syndrome [6,7,8,9]. As evidenced by these studies, plasma TMAO has become an important biomarker for determining the human health status in a variety of disease states. Recently, we reported a positive correlation between plasma TMAO and urine TMAO concentrations, either in spot urine or in 24 h urine collections [10]. These data suggest that urine TMAO can also be highly indicative of health status similar to plasma TMAO. Importantly, when compared to serum or plasma, urine is much easier to collect as it does not require venipuncture. 

To date, little has been reported on the clinical relevance of TMA due to nearly non-detectable concentrations in plasma, with the exception of patients with fish odor syndrome resulting from a deficiency in hepatic flavin monooxygenase 3 [11]. The non-detectable TMA in healthy controls may be due to the rapid oxidation of TMA in liver or excretion via urine and it is reported that more than 90% of the TMA produced from diet can be oxidized as TMAO [12,13]. Here we demonstrated that TMA can be detected in human urine, a finding not limited to patients with fish odor syndrome [14]. Previously, different methods have been used to measure urine TMA such as proton nuclear magnetic resonance (^1^H NMR), gas chromatography–mass spectrometry (GC/MS), liquid chromatography–mass spectrometry (LC/MS) and fast atom bombardment mass spectrometry [15,16,17,18,19,20]. For GC/MS, headspace and microfiber extraction were used to introduce samples. Urine TMAO can be measured simultaneously with TMA by LC/MS and fast atom bombardment mass spectrometry [14,19,21,22]. However, fast atom bombardment mass spectrometry is not widely used as an analytical instrument compared to LC/triple quadrupole MS. Additionally, in healthy humans, the average urine concentration of TMAO is about 160 times higher than that of TMA [22], suggesting that when the mass spectrometer’s TMAO signal is saturated, the TMA signal may remain undetectable. Consequently, there exists an unmet need to relatively increase the TMA recovery rate while simultaneously decreasing the TMAO recovery rate before transferring to a mass spectrometry vial for LC/MS. Such a method would promote favorable conditions that allow for the quantification of urine TMA and TMAO with a large concentration range as is observed in human urine samples. In this study, we developed a method by the hexane/butanol extraction of TMAO and TMA under alkaline pH in urine to decrease the relative recovery rate of TMAO to TMA to attain this aim.

## 2. Results

### 2.1. Validation of the LC/MS/MS Quantitation of TMA and TMAO in Urine

Plasma/serum and urine TMAO can be simultaneously measured with other TMA-related metabolites such as choline, carnitine, γ-butyrobetaine and crotonobetaine by LC/MS/MS after precipitation of protein with methanol [10]. Using the same method, urinary TMAO yielded a S/N > 10, yet the TMA peak was unacceptable due to a high baseline and multiple peaks in the same channel (Figure 1A). Adjusting the injection volume failed to yield an acceptable TMA peak since the background signal changed accordingly, which may be related to other compounds containing a TMA moiety in urine. Furthermore, an increased injection volume can lead to TMAO signal saturation, which is not conducive to linearity throughout the detectable range. Although we can monitor another daughter ion of TMAO with a lower sensitivity or change the MS parameters to monitor ions at a sub-optimal status to avoid signal saturation, the larger injection volume required to do so would lead to a contamination of the MS source and quadrupoles thus adding to the cost of instrument maintenance. Alternatively, we utilized a method to extract the TMAO and TMA with hexane/butanol under alkaline pH followed by acidification to transfer to the aqueous phase. This approach allowed for the simultaneous measurement of urine TMAO and TMA by LC/MS/MS (Figure 1B), where we observed acceptable peaks for both TMA and TMAO. 

In an effort to optimize the recovery rate, TMA, d9-TMA, TMAO, and d9-TMAO, at a concentration of 100 μM each, were extracted by the procedure described above with variable volumes of butanol and variable concentrations of 1 mL NaOH (Figure 2). Without the addition of NaOH, TMA and d9-TMA, recovery was not observed. Varying the concentration of NaOH from 0.1 M to 0.5 M had little effect on the recovery rate whereas the addition of butanol greatly improved the recovery rate of TMA and d9-TMA. For TMAO and d9-TMAO, the addition of 0.2 M NaOH significantly improved the recovery rate as did the addition of 1 mL butanol. Based on the recovery curves for TMA, d9-TMA, TMAO, and d9-TMAO, we observed that the addition of butanol uniformly improved the recovery rate. However, due to the high boiling point of butanol (bp 116 °C), butanol may not be compatible with mass spectrometry. Moreover, the relatively higher solubility of butanol in water compared to hexane may lead to excessively high pressures in the subsequent liquid chromatography (LC) system. Nevertheless, we prepared human urine samples by adding 1 mL butanol and 1 mL 0.5 M NaOH for the quantitation of urine TMA and TMAO. Under such conditions, the recovery rates of TMA, d9-TMA, TMAO, and d9-TMAO were 75.3%, 78.5%, 0.50%, and 0.53%, respectively. The recovery rate ratio of TMA to TMAO was 150, which normalized the concentration difference between endogenous levels of urinary TMAO and TMA. The end result was an extracted urine sample ready for electrospray ionization (ESI) LC/MS analysis with TMA and TMAO concentrations of the same magnitude.

The linearity of the standard curve, which was a plot of the peak area ratio of TMA to d9-TMA and TMAO to d9-TMAO, versus TMA and TMAO concentrations, respectively, was related to the volume of butanol and NaOH added. Furthermore, the linearity was more sensitive to butanol than NaOH (Figure 3A–F)**.** The standard curves for TMA were uniformly linear with r^2^ > 0.99 except when extracting with 1 mL butanol and 1 mL 0.2 N NaOH (r^2^ = 0.9897). The slopes were very different with respect to the added butanol volume and NaOH concentration with a coefficient of variation (CV) of 13.8% if the effects of butanol and NaOH were ignored (Figure 3A–C). For TMAO (Figure 3D–F), the addition of 0.2 mL butanol or no butanol did not yield a linear curve (r^2^ < 0.99), which was due to the low recovery rate. The linearity was largely affected by the addition of 1 mL butanol yet the addition of NaOH at different concentrations did not significantly change the linearity of either TMA or TMAO—suggesting that butanol showed discrimination in the recovery rates between native TMA or TMAO and deuterium-labeled TMA/TMAO, respectively. This finding was further confirmed by the recovery rate comparison (Figure 2). Isotope-labeled standard usually has the same physicochemical property as non-labeled standard, which is widely used to quantify metabolites by mass spectrometry with very high accuracy. Here, we observed a modest difference between the TMA and TMAO and their respective deuterium-labeled standards, d9-TMA and d9-TMAO. Compared to deuterium-labeled compounds, ^13^C or ^15^N-labeled compounds should be more similar to the naturally abundant compound in terms of their physicochemical properties. In order to confirm this, we prepared standard curves using [^13^C_3_,^15^N]TMA and [^13^C_3_]TMAO as internal standards. Of note, the slope difference with the addition of the different volume of butanol was still observed (Figure 4), but with a trivial CV% of 2.9% for the TMA and 6.0% for the TMAO. Consequently, the effects of butanol volume were ignored during the quantitation as the CV%s were drastically smaller than the CV%s observed using d9-TMA and d9-TMAO as internal standards. However, due to cost, the deuterium-labeled compounds d9-TMA and d9-TMAO were used as internal standards in the subsequent study while strictly controlling the volume of butanol and the concentration of NaOH added. Using different known concentrations of TMA and TMAO standards to spike urine, the slopes of the spike-in standard curves had CV%s of 0.8% and 1.2% for TMA and TMAO, respectively (Figure 5). The TMA slope was 9.2% different from the non-spike-in standard curve and the difference was less than 1.5% for the TMAO (Figure 5). As such, the use of a spike-in standard curve represented a reliable method to measure the TMA in urine whereas either a spike-in standard curve or a non-spike-in curve may be used to measure TMAO in urine. The TMA and TMAO standard curves had a wide range of linearity, with concentrations extending to 100 μM for the TMA and 10,000 μM for the TMAO; all the while maintaining slopes of linearity with less than 3% variation (data not shown).

The detection limit is related to the instrument used and the injection volume. For the Shimadzu 8050 LC/MS with a 2 μL injection, the detection limit for TMA was 0.26 μM and 0.57 μM for TMAO. The limit of quantitation for TMA was 0.40 μM and 5.0 μM for TMAO. The accuracy of TMA and TMAO measurements are listed in Table 1, where we can see that the accuracy is very close to 100%, with 96.2% and 98.4% accuracy maintained at a concentration very close to the lower limit of quantitation for TMA and TMAO, respectively. 

To examine assay precision, urine TMA and TMAO were measured in six different human urine samples over a span of more than 10 days. Shown in Table 2, we can see that the intra-day CV%s for TMA and TMAO are less than 4.0% and 9.0%, respectively, and the inter-day CV%s are higher than intra-day CV%s. Only the urine sample with a relatively low TMA concentration of 0.99 μM gave a CV% greater than 15% while all the other samples gave CV%s less than 15% for both TMA and TMAO. 

Compounds with a TMA structural moiety can serve as substrates for TMA production by gut microbial enzymatic cleavage [2,10,23,24]. In order to test whether these substrates can be spontaneously degraded to form TMA during sample processing in our method, we spiked 100 µM d9-choline, d9-betaine, d9-TMAO, d9-carnitine, d9-γ-butyrobetaine, and d9-crotonobetaine into five randomly selected human urine samples (500 µL each), separately, followed by the addition of 2 mL hexane/1 mL butanol and 1 mL 0.5 M NaOH. After being vortexed and spun down, the hexane layer was acidified with 0.2 mL 0.2 M formic acid to collect the aqueous phase. By LC/MS/MS, we failed to detect any d9-TMA, suggesting that no artificial production of TMA occurred during sample processing. 

### 2.2. Normal Range of Urinary TMA and TMAO in Healthy Subjects

To determine the normal range of urinary TMA and TMAO levels, apparently healthy human subjects (n = 29) undergoing routine healthy screens in the community were examined. Table 3 shows the distribution of urinary TMA and TMAO. Urinary TMAO has a relatively larger concentration range compared to TMA, with a ratio of maximum to minimum concentration of TMAO at 22 and 6 for TMA. The minimum TMA and TMAO concentrations were 0.70 and 52.0 μM, respectively, which were above the LLOQs. Shown here, we can see that urine TMAO concentrations were much greater than TMA, with half of the human samples containing TMAO at a concentration >200 fold higher than TMA. Furthermore, urinary TMAO and TMA were highly correlated to each other (r^2^ = 0.27, *p* = 0.004).

### 2.3. Comparison of the Measured TMAO Results by Two Different Methods 

Urine TMAO can be directly measured by mixing with the internal standard dissolved in methanol. After precipitation and removal of protein by centrifugation, the supernatant can be injected into an LC/MS system as described above. We compared this method with the hexane/butanol extraction method for quantitation of urine TMAO. Results are shown in Figure 6, where we observed a linearity between the concentration of urine TMAO acquired by the two different methods (r^2^ = 0.948, slope = 1.094, CV%=14.3). These data suggested that the results determined by either method are both comparable and reliable.

### 2.4. Stability of Urine TMA

Urine is a collection of metabolic waste containing commensal genitourinary tract microbes responsible for maintaining the health of the urinary tract in vivo [26]. Given that the molecular TMA precursors, choline, betaine, carnitine, γ-butyrobetaine and crotonobetaine are abundant in urine [10], TMA concentrations may increase over time due to metabolism by microbial enzymes. We selected two urine samples and monitored the concentration of TMA during storage at different time points and temperatures. We reported urinary TMA concentration to be stable after one week of storage at −80 °C, −20 °C and 4 °C since the concentration differed from baseline by less than 15%. However, after prolonged storage up to five years, a dramatic increase in urine TMA was observed (Figure 7). The storage at room temperature, 37 °C and 60 °C will lead to a rapid increase in urine TMA concentration.

Additionally, due to its gaseous properties, TMA may be volatile during storage. In order to confirm this, we tested the stability of a TMA stock solution in sterile water with concentrations ranging from 10 to 40 mM during storage at room temperature while maintaining a pH close to 7.0. After two and three months of storage, there was no observable difference in the TMA concentration as evidenced by a deviation of less than 15% from baseline and a CV% of 12.0% across three different timepoints for the seven stock solutions. These data suggested that TMA in sterile water is stable and the gaseous properties of TMA do not affect its stability in water during storage at a neutral pH (Figure 8). However, the stock solution shown here has a concentration ~1000 times higher than that what is found in human urine samples. Consequently, the possibility persists that the volatility of TMA may still contribute to instability at physiological concentrations. In some papers, the acidification of urine was suggested for storage for TMA quantification [21]. In our study, we compared the urine TMA concentration changes between the samples with 60 mM HCl added versus samples with no HCl added. We determined that HCl can significantly attenuate the increase in urine TMA (*p* < 0.05) during storage for three months or five years either at −80 °C or room temperature (Figure 9). Granted, compared to baseline, urine with HCl added still showed an increase in TMA concentration after 5 years, suggesting that urinary TMA is continually generated during storage but acidification can delay this spontaneous production. The observed increase in urinary TMA during storage may be related to the presence of genitourinary bacteria. In order to test this, we sterilized urine using a syringe filter (0.22 µm, Millipore) and incubated the sterilized urine with deuterium-labeled TMA-containing compounds. Here we reported that d9-TMAO can gradually be metabolized to d9-TMA in non-sterilized urine and that sterilized urine is less conducive to the catabolism of TMAO to TMA (Figure 10). These data suggested that the bacterial TMAO reductases may contribute to the increase in TMA observed during storage. 

## 3. Discussion

Herein we presented an LC/MS/MS method for the simultaneous measurement of TMA and TMAO in urine with high accuracy and precision. Hexane/butanol extraction under alkaline condition and subsequent transfer to the aqueous phase after acidification resulted in a sample suitable for mass spectrometric analysis. Furthermore, this method of sample preparation ensured the MS source free of contamination and shortened the data acquisition window due to the removal of compounds with a structural formula containing a TMA moiety. The different recovery rates of TMAO and TMA by sample preparation offset the concentration difference while running LC/MS/MS and the linearity of urinary TMAO measurement can be extended to more than 10 mM from 5 μM. Moreover, it was observed that the accuracy of urine TMA and TMAO measurement was largely affected by the volume of butanol added to the extraction system. This is a high-throughput method, where prepared samples can be injected using an automated sampling system and the LC/MS can be done in 9.5 min per injection. Using a two LC column system as we reported previously [25], a single injection can be completed in 5 min. 

Urinary TMA is not stable during storage and the concentration will increase with time and storage temperature. The acidification of urine was expected to decrease the volatility of TMA but meanwhile it was determined to play a role in the inhibition of TMA production from other TMA containing compounds, such as TMAO. TMAO may spontaneously be metabolized to TMA and urogenital bacterial flora may speed up this conversion process. Compared with TMA, urinary TMAO concentrations are much higher, thus the slow conversion will not lead to a significant change in urine TMAO concentration. 

The highly positive correlation between urine TMAO and plasma TMAO [10], and the clinical relevance of TMAO [1,5,6,27], confer the practical use of urine TMAO measurement. Monitoring urine TMAO can serve as a biomarker for diet–gut microbiota interactions. 

The demonstration of the involvement of TMAO in cardiovascular disease [3,5] suggests that inhibition in this metaorganismal pathway can be beneficial for cardiovascular health [28,29]. From dietary TMA precursors to TMAO, there are two steps: the first step is gut microbial enzymatic cleavage to produce TMA, which is the rate-limiting step. As such, the inhibition of this first step becomes a therapeutic strategy to attenuate atherosclerosis and thrombosis. Using the present method for measuring urine TMA, we can generate a high-throughput screen of inhibitors to specific TMA-containing compounds by incubation with purified bacterial enzymes, bacterial lysate, intact bacterial cells, or human fecal samples. The hexane/butanol extraction under alkaline pH can also be used to measure the TMA content in fecal samples after homogenization in cold water. 

## 4. Materials and Methods

### 4.1. Reagents

Deuterated trimethylamine-*N*-oxide (d9-TMAO, DLM-4779-1) and TMA (d9-TMA, DLM-1817-5) were purchased from Cambridge Isotope Laboratories (Tewkesbury, MA, USA). [^13^C_3_]Trimethylamine *N*-oxide was purchased from IsoSciences (Ambler, PA, USA). All other reagents were purchased from either Sigma-Aldrich (St. Louis, MO, USA) or Fisher Scientific Chemicals (Waltham, MA, USA) unless otherwise stated.

### 4.2. Research Subjects

Urine samples were collected from subjects undergoing community health screens. All subjects gave written informed consent and the Institutional Review Board of the Cleveland Clinic approved all study protocols.

### 4.3. Sample Processing

500 μL urine was aliquoted in a 13 × 100 mm glass tube, and 20 μL isotope labeled TMA and TMAO, d9-TMA and d9-TMAO, were added with fixed concentrations. Then, the urine TMA and TMAO were extracted in the organic phase by adding 2 mL hexane and different volumes of butanol and 1 mL NaOH of varying concentrations. After being vortexed for 1 min and spun down at 2500 g, 4 °C for 10 min, the hexane layer was collected and transferred to a 12 × 75 mm glass tube with 0.2 mL 0.2 N formic acid pre-loaded. After being vortexed for 1 min and spun down at 2500 g, 4 °C for 5 min, the aqueous phase was collected in a mass spec vial (Agilent Technologies, Santa Clara, CA, USA, Part: 5182-0714) with plastic insert (P. J. Cobert Association, St. Louis, MO, USA, Cat. No. 95172) and a Teflon cap (Agilent Technologies, Part: 5182-0717) for liquid chromatography–tandem mass spectrometry (LC/MS/MS) analysis. Different concentrations of TMA and TMAO replacing urine or spiked to urine samples were used to prepare calibration curves in parallel.

For comparison, an alternative method for urine TMAO quantitation is to dilute urine 20 times with water, then aliquot 20 μL diluted urine to a 1.5 mL Eppendorf tube, mixed with 80 μL 10 μM d9-TMAO in methanol. Then, vortex for 1 min and spin down at 20,000 g, 4 °C for 10 min and transfer the supernatant to a mass spectrometry (MS) vial for LC/MS/MS analysis. Then, 20 μL aliquots of varying concentrations (0−100 μM) of TMAO were mixed with 80 μL 10 μM d9-TMAO in methanol to prepare a calibration curve.

### 4.4. LC/MS/MS

Prepared MS samples were analyzed by injection onto a silica column (2 × 150 mm, 5 mm Luna silica; Cat. No: 00F-4274-B0, Phenomenex, Torrance, CA, USA) at a flow rate of 0.3 mL.min^−1^ using a 2 LC-20AD Shimadzu pump system, a SIL-HTC autosampler interfaced with a QTRAP5500 mass spectrometer (AB SCIEX, Framingham, MA, USA) or a Shimadzu 8050 mass spectrometer or a Vanquish LC system interfaced with a Thermo Quantiva mass spectrometer. A discontinuous gradient was generated to resolve the analytes by mixing solvent A (0.2% formic acid in water) with solvent B (0.2% formic acid in methanol) at different ratios starting at 0% B for 2 min then linearly to 15% within 4 min, then linearly to 100% B within 0.5 min, then held for 3 min and then back to 0% B. TMA, TMAO, d9-TMA, d9-TMAO, [^13^C_3_,^15^N] TMA, [^13^C_3_]TMAO were monitored using electrospray ionization (ESI) in positive-ion mode with multiple reaction monitoring (MRM) of precursor and characteristic product ion transitions of m/z 60 → 44 amu, 76 → 58 amu, 69 → 49 amu, 85 → 66 amu, 64 → 47 amu and 79 → 61 amu, respectively. The parameters for the ion monitoring were optimized in individual mass spectrometers. The internal standards d_9_-TMA (or [^13^C_3_,^15^N] TMA), d_9_-TMAO (or [^13^C_3_] TMAO) were used for quantification of TMA and TMAO, respectively. 

### 4.5. Precision, Accuracy, Limit of Quantitation and Linearity

Four replicates were performed on a single day to establish the intra-day coefficient of variation (CV) for 6 different urine samples with different concentrations of TMA and TMAO. The inter-day CV was determined by assaying aliquots of these samples daily over a span of more than 10 days. Carry-over between injections was not observed. Accuracy was expressed as the ratio of the TMA, the TMAO concentration measured to the TMA, the TMAO concentration added to water undergoing the hexane/butanol extraction. The lower limit of detection (LLOD) was calculated via calibration approach using Equation C [30]. The lower limit of quantification (LLOQ) was estimated from calibration standards as the lowest concentration with a calculated deviation ≤ 30% based on the calibration curve [30]. To determine assay linearity, a standard curve over the 0.01–100 µM, 10–3000 μM concentration range for TMA and TMAO, respectively, was checked for linearity by linear regression fit. The linear range was defined as the region of the standard curve where the difference between the calculated TMA or TMAO concentration and the standard TMA or TMAO concentration was less than 15%.

## Figures and Tables

**Figure 1 molecules-25-01862-f001:**
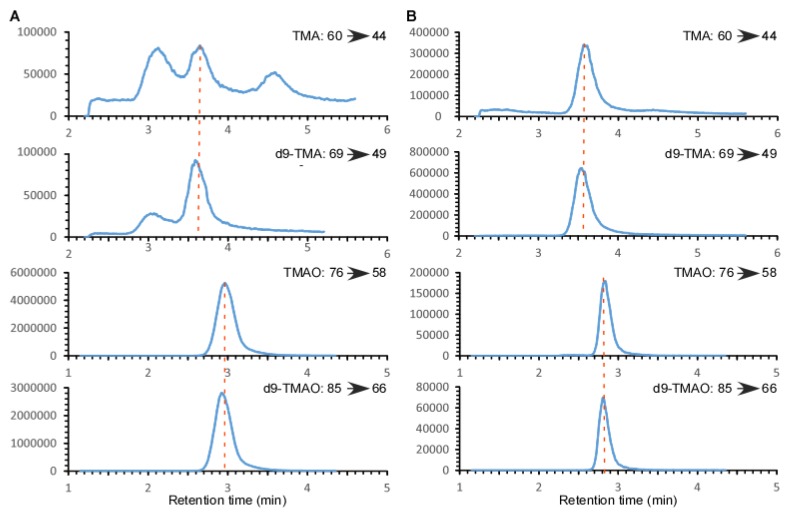
Extracted-ion LC chromatograms from the multiple reaction monitoring (MRM) in positive-ion mode of trimethylamine (TMA), deuterated trimethylamine (d9-TMA), trimethylamine-*N*-oxide (TMAO) and deuterated trimethylamine-*N*-oxide (d9-TMAO) in a typical human urine sample by two different processing methods. (**A**): Mixed with 4 volumes of methanol and internal standards d9-TMA and d9-TMAO; (**B**): Extracted with hexane/butanol after spiking with internal standards d9-TMA and d9-TMAO, under alkaline pH followed by acidification to transfer to the aqueous phase. Then, 5 μL sample was injected onto a Silica column followed by a solvent elution as described in the methods. The eluate was monitored for TMA, TMAO, d9-TMA and d9-TMAO in a Thermo Quantiva mass spectrometer. The precursor-to-product transitions were m/z 60 → 44, m/z 69 → 49, m/z 76 → 58 and m/z 85 → 66 for TMA, d9-TMA, TMAO and d9-TMAO, respectively. The concentrations of TMAO and TMA in this urine sample were measured by the hexane/butanol extraction method as 314 μM and 2.37 μM, respectively. The data were acquired in Thermo Quantiva mass spectrometer with a Vanquish auto-sampler and LC pump system.

**Figure 2 molecules-25-01862-f002:**
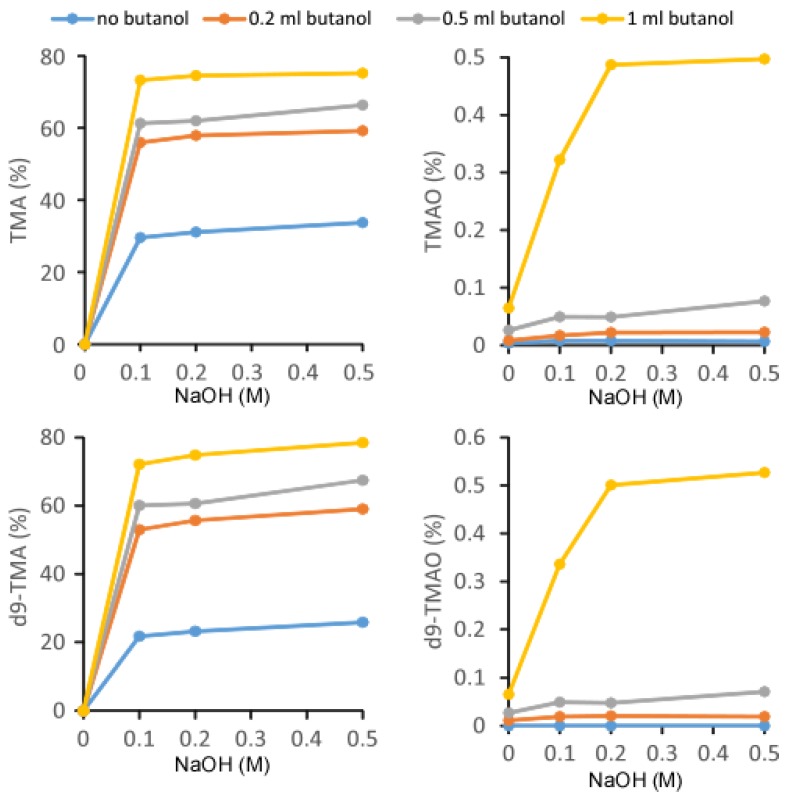
Recovery rates of TMA, TMAO, d9-TMA and d9-TMAO. Here, 500 μL of 40 μM (TMA + TMAO) was mixed with 20 μL of 1 mM (d9-TMA + d9-TMAO) and extracted with 2 mL hexane and different volumes of butanol in the presence of 1 mL NaOH at varying concentrations. This step was followed by the transfer to the aqueous phase by adding 0.2 mL of 0.2 N formic acid. Then, 5 μL was injected into the liquid chromatography–mass spectrometry (LC/MS). The recovery rate (%) was calculated by the peak area of the standard after extraction preparation divided by the peak area of the standard, 100 μM (TMA + TMAO + d9-TMA + d9-TMAO), without undergoing extraction. Data were acquired in a Thermo Quantiva mass spectrometer with a Vanquish auto-sampler and a LC pump system.

**Figure 3 molecules-25-01862-f003:**
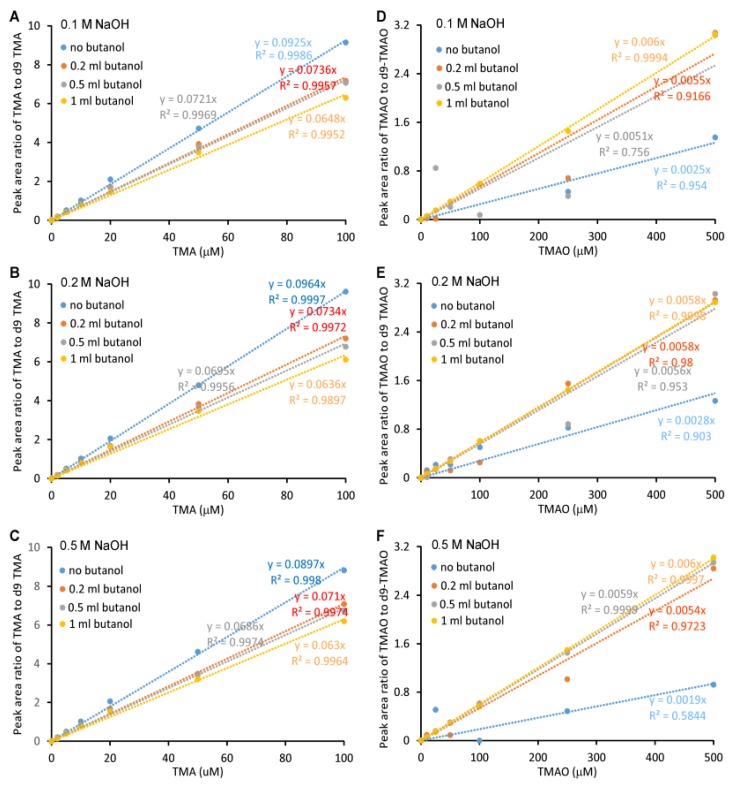
Standard curves for the Liquid Chromatography Electrospray Ionization Tandem Mass Spectrometric (LC/ESI/MS/MS) analysis of TMA and TMAO. Here, 500 μL of varying concentrations of TMA and TMAO were mixed with 20 μL of 1 mM (d9-TMA + d9-TMAO) and extracted with 2 mL hexane and different volumes of butanol in the presence of 1 mL NaOH at varying concentrations ((**A**, **D**), 0.1 M; (**B**, **E**) 0.2 M; (**C**, **F**), 0.5 M). Then, analytes were transferred to the aqueous phase by adding 0.2 mL of 0.2 M formic acid. Curves were plotted as the peak area ratio of TMA to d9-TMA (**A**–**C**) and TMAO to d9-TMAO (**D**–**F**) versus concentration. Data were acquired in a QTRAP5500 mass spectrometer with a Shimadzu auto-sampler and a LC pump system.

**Figure 4 molecules-25-01862-f004:**
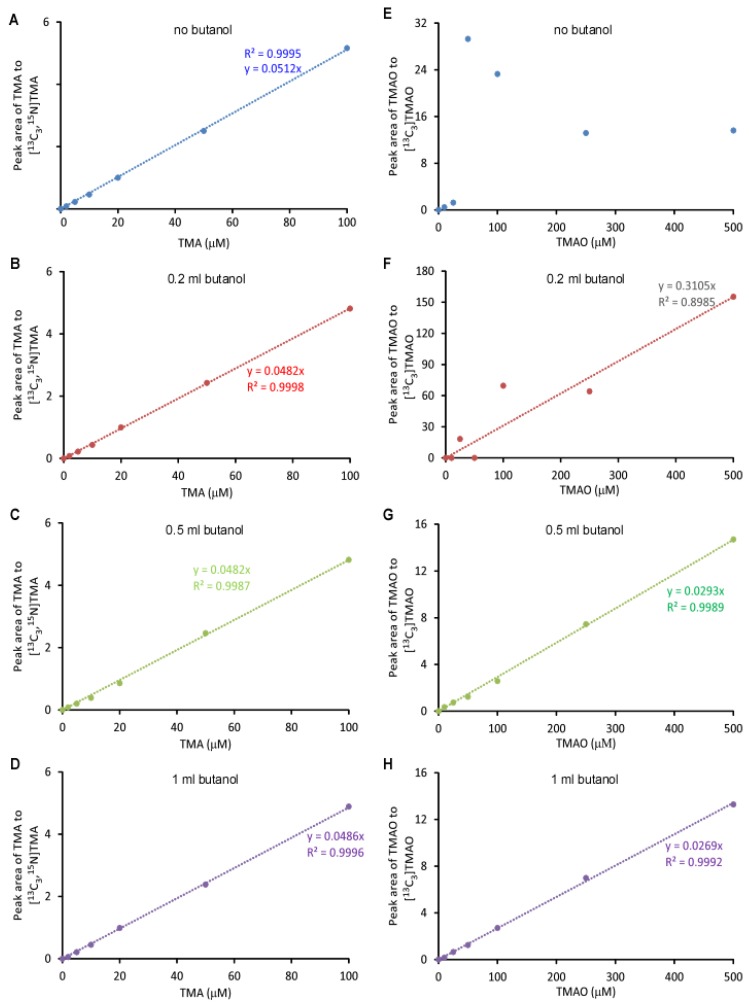
Standard curves for the LC/ESI/MS/MS analysis of TMA and TMAO with [^13^C_3_,^15^N]TMA and [^13^C_3_]TMAO as internal standards, respectively. Here, 500 μL of varying concentrations of TMA and TMAO were mixed with 20 μL of 1 mM [^13^C_3_, ^15^N]TMA and 1 mM [^13^C_3_]TMAO and extracted with 2 mL hexane and varying volumes of butanol (no butanol (**A**,**E**), 0.2 mL butanol (**B**,**F**), 0.5 mL butanol (**C**,**G**), 1 mL butanol (**D**,**H**)) in the presence of 1 mL 0.5 M NaOH. Then, analytes were transferred to the aqueous phase by adding 0.2 mL of 0.2 M formic acid. The analysis was performed using electrospray ionization in positive-ion mode with multiple reaction monitoring of precursor and characteristic product ions. The transitions monitored were mass-to-charge ratio (*m*/*z*): *m*/*z* 60 → 44, *m*/*z* 64 → 47, *m*/*z* 76 → 58 and *m*/*z* 79 → 61 for TMA, [^13^C_3_, ^15^N]TMA, TMAO and [^13^C_3_]TMAO, respectively. Curves were plotted as the peak area ratio of TMA to [^13^C_3_, ^15^N]TMA (**A**–**D**) and TMAO to [^13^C] TMAO (**E**–**H**) versus concentration. Data were acquired in a Thermo Quantiva interfaced with a Vanquish LC system with 0.5 μL injection to column.

**Figure 5 molecules-25-01862-f005:**
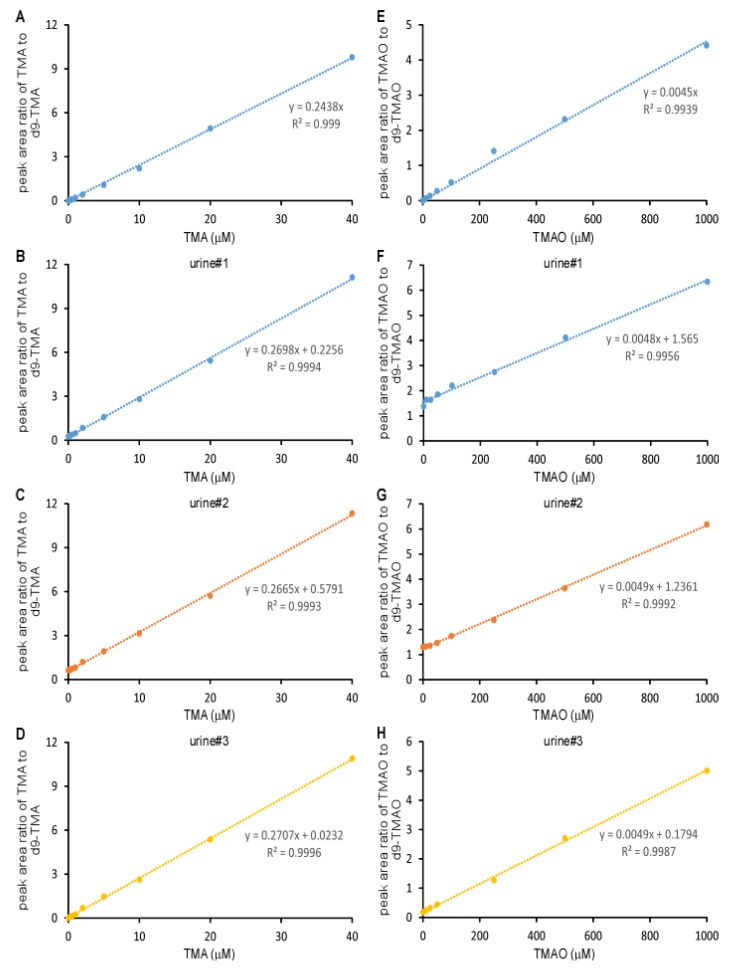
Standard curves for the LC/ESI/MS/MS analysis of the TMA and TMAO spiked into urine or non-urine control with d9-TMA and d9-TMAO as internal standards, respectively. Varying concentrations of TMA and TMAO standards were spiked into 3 urine samples or water. Then, 500 μL each was mixed with 20 μL of 1 mM d9-TMA and 1 mM d9-TMAO, extracted with 2 mL hexane and 1 mL butanol in the presence of 1 mL 0.5 N NaOH, and transferred to the aqueous phase by adding 0.2 mL 0.2 N formic acid. Curves were plotted as the peak area ratio of TMA to d9-TMA (**A**–**D**) and TMAO to d9-TMAO (**E**–**H**) versus concentration. Data were acquired in a Shimadzu 8050 LC/MS/MS with 2 μL sample injected into column.

**Figure 6 molecules-25-01862-f006:**
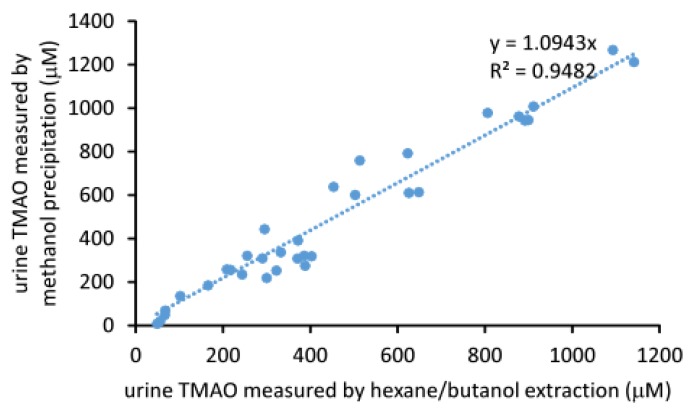
Correlation between the two different methods for the measurement of urine TMAO. Hexane/butanol extraction performed as described in Figure 5. For the methanol precipitation method, urine was diluted 20 times first, then mixed with 4 volumes of methanol containing 10 μM d9-TMAO as the internal standard following the method for the determination of the plasma TMAO [25].

**Figure 7 molecules-25-01862-f007:**
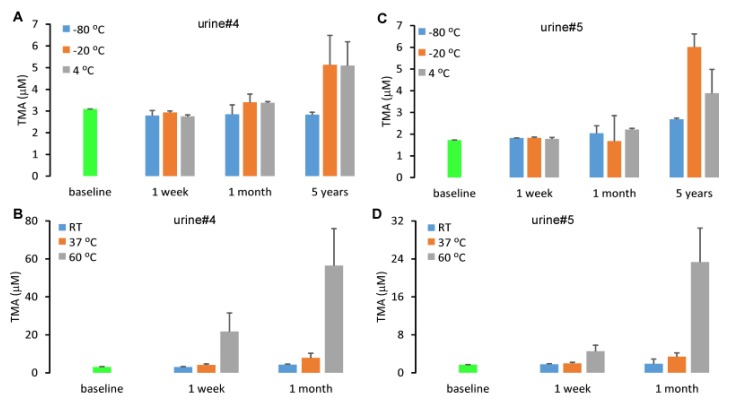
Urine TMA concentration changes during storage. Two urine samples were stored at different temperatures, −80 °C, −20 °C 4 °C (**A**,**C**), room temperature (RT), 37 °C and 60 °C for varying times (**B**,**D**). Urine TMA was extracted with hexane/butanol under alkaline pH with d9-TMA as an internal standard and quantified by LC/MS/MS. Data were presented as mean ± SD from two replicates.

**Figure 8 molecules-25-01862-f008:**
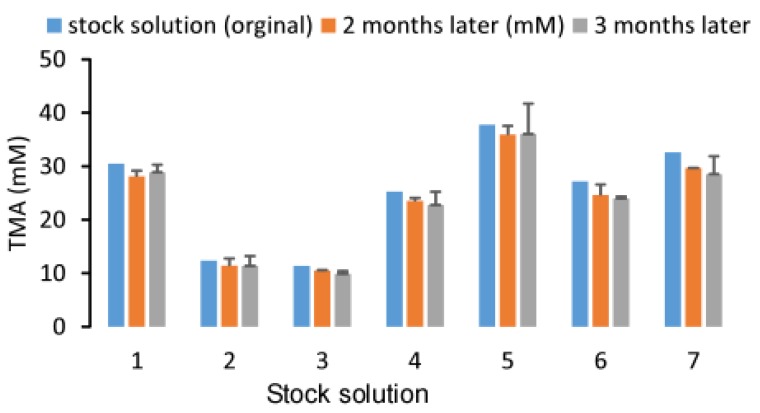
Stability of the TMA stock solution during storage at room temperature. Varying concentrations of TMA standards in sterile water were stored in glass vials and kept at room temperature. The concentrations after 2 and 3 months of storage were measured by LC/MS/MS. Each stock solution was diluted to 10–50 μM and calibrated with standard curves after spiking-in a fixed amount of d9-TMA as an internal standard. Data were presented as mean±SD from three replicates.

**Figure 9 molecules-25-01862-f009:**
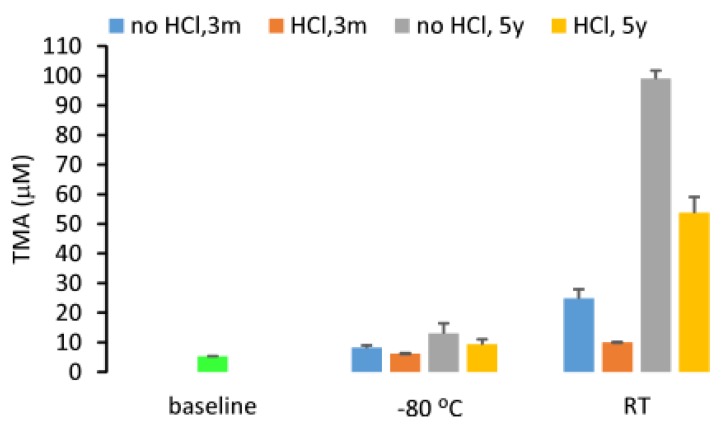
Urine TMA concentration changes during storage after acidification at −80 °C or at room temperature (RT). The HCl added to urine has a final concentration of 60 mM. Data were presented as mean±SD from two replicates.

**Figure 10 molecules-25-01862-f010:**
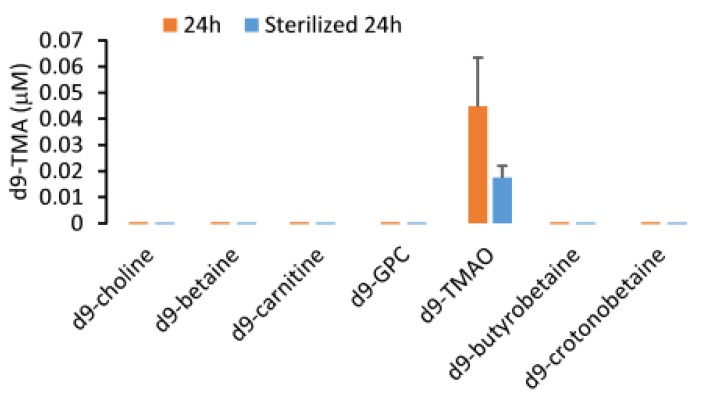
The d9-TMA monitoring in urine after incubation with different stable isotope-labeled compounds containing a TMA (d9-TMA) moiety. Here, 100 μM d9-choline, d9-betaine, d9-carnitine, d9-glycerophosphocholine (GPC), d9-TMAO, d9-butyrobetaine and d9-crotonobetaine were separately spiked into a pooled urine sample randomly selected from 5 human subjects. Urine filtered through a 0.22 μm filter was used as sterilized urine. Then, 500 μL of the non-sterilized urine or sterilized urine were put in glass tube and incubated at 37 °C for 24 h for determination of d9-TMA. In addition, 20 μL of 100 μM [^13^C_3_,^15^N]TMA was added to the urine sample as the internal standard followed by the addition of 2 mL hexane/1 mL butanol and 1 mL of 0.5 N NaOH for extraction. The extracted TMA was transferred to the aqueous phase by adding 0.2 mL of 0.2 M formic acid. Serial dilution of the d9-TMA in 500 μL water mixed with 20 μL of 100 μM [^13^C_3_,^15^N]TMA followed by the same procedure that was used to determine the concentration of TMA in urine was used to prepare a calibration curve by LC/MS/MS. The precursor-to-product transitions in positive MRM mode monitoring by mass spectrometry were m/z 69 → 49 and *m*/*z* 64 → 47 for d9-TMA and [^13^C_3_,^15^N]TMA, respectively. Data were presented as mean±SD from two replicates.

**Table 1 molecules-25-01862-t001:** Characteristics of the method for the TMAO and the TMA determination by LC/MS/MS.

Characteristic	TMA		TMAO	
LLOD	0.26 µM		0.57 µM	
LLOQ	0.40 µM		5.0 µM	
ULOQ	>50 µM		>10 mM	
Accuracy (%)	0.5 µM	96.2	10 µM	98.4
	2 µM	97.0	50 µM	102.8
	5 µM	101.1	100 µM	102.1
	10 µM	102.0	250 µM	102.1
	20 µM	102.3	500 µM	100.6
	40 µM	101.3	2000 µM	101.3

Lower limit of detection (LLOD) and Lower limit of quantitation (LLOQ) were estimated by Calibration Approach as described in methods. Upper limit of quantitation (ULOQ) = greater than highest standard investigated. Accuracy = ratio of the TMA and TMAO concentrations measured to the TMA and TMAO concentrations added to water followed by the same procedure as the urine samples for the TMAO and TMA measurement, respectively.

**Table 2 molecules-25-01862-t002:** Precision of the TMA and the TMAO concentrations (in µM) measured in 6 urine samples with different levels over a span of more than 10 days.

Urine	TMA			TMAO		
	Mean ± SD	Intraday CV%	Interday CV%	Mean ± SD	Intraday CV%	Interday CV%
1	0.99 ± 0.09	3.4	17.6	305.6 ± 22.7	5.2	12.4
2	2.41 ± 0.10	2.2	8.6	236.8 ± 15.9	5.1	10.5
3	2.28 ± 0.14	2.1	12.3	303.3 ± 11.8	2.9	6.1
4	4.60 ± 0.18	2.5	7.0	393.3 ± 30.3	8.5	2.3
5	19.54 ± 0.65	2.1	6.2	825.2 ± 68.1	7.8	10.1
6	9.96 ± 0.34	2.5	5.9	1215.2 ± 47.3	2.6	7.3

The calculated mean, SD, intraday CV% and interday CV% are given. For each of the 6 different urine samples, a cluster of 4 determinations is presented as they were run on four separate days. CV, coefficient of variance.

**Table 3 molecules-25-01862-t003:** Quantile distribution of urinary TMA and TMAO.

	0%	25%	50%	75%	100%
TMA (μM)	0.70	1.37	2.07	2.62	4.39
TMAO (μM)	52.0	243.7	379.0	648.1	1141.0
TMAO/TMA (mol/mol)	66	132	208	315	506
TMA/creatinine (mmol/mol)	0.05	0.09	0.13	0.23	1.43
TMAO/creatinine (mmol/mol)	11.3	22.5	33.8	45.6	106.4

Twenty-nine healthy human subjects’ urine samples were collected to measure TMA and TMAO. Urine creatinine was measured by LC/MS following the protocol as reported previously [10].

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
