# Peer review of "Simultaneous Measurement of Urinary Trimethylamine (TMA) and Trimethylamine *N*-Oxide (TMAO) by Liquid Chromatography–Mass Spectrometry"

_molecules, 2020, doi:10.3390/molecules25081862_

Round 1

Reviewer 1 Report

The manuscript presents an interesting application of liquidchromatography-mass spectrometry for simultaneous measurement ofurinary trimethylamine (TMA) and trimethylamineN-oxide (TMAO).

The paper is well presented, and the experimental methodology well planned. The interesting results include developed a method by hexane/butanolextractionofTMAO andTMAunderalkalinepHinurinetodecreasetherelativerecoveryrateofTMAOtoTMA.

However, before the manuscript can be considered suitable for publication in the Molecules, some minor revisions are needed.

In particular, the following points should be clarified:

Please better explain how you choose criteria for the condition to check the stability of urine TMA (storage at different time points and temperatures, figures 7, 8 and 9). Then, how do they explain the changes in stability. As a general comment, the authors should have major care in checking the manuscript: there a lot of typographic and grammatical errors throughout the text.

Author Response

The manuscript presents an interesting application of liquid chromatography-mass spectrometry for simultaneous measurement of urinary trimethylamine (TMA) and trimethylamine N-oxide (TMAO).

The paper is well presented, and the experimental methodology well planned. The interesting results include developed a method by hexane/butanol extraction of TMAO and TMA under alkaline pH in urine to decrease the relative recovery rate of TMAO to TMA.

However, before the manuscript can be considered suitable for publication in the Molecules, some minor revisions are needed.

In particular, the following points should be clarified:

Response: Thank the reviewer for the positive comments and the helpful modification suggestions. We are now addressing his/her concerns one by one and meanwhile we modified the manuscript based on his/her comments.

Point 1: Please better explain how you choose criteria for the condition to check the stability of urine TMA (storage at different time points and temperatures, figures 7, 8 and 9). Then, how do they explain the changes in stability.

Response 1: We have calculated the difference of urine TMA concentrations from baseline and defined a difference of less than 15% as stable. Urine TMA stability changes during storage may be related to the bacterium TMAO reductase metabolism of TMAO to TMA or spontaneous conversion of TMO to TMA.

Point 2: As a general comment, the authors should have major care in checking the manuscript: there a lot of typographic and grammatical errors throughout the text.

Response 2: We have checked the manuscript and tried our best to correct any typographic and grammatical error. 

Reviewer 2 Report

The manuscript of Jia et al. reports on an LC-MS based analytical method for the determination of TMA and TMA-Oxid in urine. I  consider the manuscript as clearly above average, as the authors have taken different urine samples for method validation (following e.g. the European Medicines Agency guide), which is an issue neglected by many other methods. However, there are some points left that need clarification in a revision (see below). In addition, I feel that the advantage of the proposed method (extraction to an organic solvent in alkaline conditions, back extraction in acidic conditions) could be pointed out more clearly by comparing the absolute response of urine spiked at a given concentration level for characterization of matrix effects (although it is true that the use of isotope labelled internal standards is a viable way for compensating those effects).

Section 2.4 There seems to be only one MS/MS transition per analyte, while state of the art are two transitions. DO the investigated analytes yiled only one product ion?

Section 2.5 The S/N ratio has recently been discouraged for determination of LOD/LOQ by the European Reference Labs (Guidance document on the estimation of LOD and LOQ for measurements in the field of contaminants in feed and food. EUR 28099 EN. http://publications.jrc.ec.europa.eu/repository/bitstream/JRC102946/eur%2028099%20en_lod%20loq%20guidance%20document.pdf). Although not legally binding for bioanalytiucal methods, the authors should consider to use another criterion.

Author Response

The manuscript of Jia et al. reports on an LC-MS based analytical method for the determination of TMA and TMA-Oxid in urine. I  consider the manuscript as clearly above average, as the authors have taken different urine samples for method validation (following e.g. the European Medicines Agency guide), which is an issue neglected by many other methods. However, there are some points left that need clarification in a revision (see below). In addition, I feel that the advantage of the proposed method (extraction to an organic solvent in alkaline conditions, back extraction in acidic conditions) could be pointed out more clearly by comparing the absolute response of urine spiked at a given concentration level for characterization of matrix effects (although it is true that the use of isotope labelled internal standards is a viable way for compensating those effects).

Response: Thank the reviewer for the positive comments and the helpful modification suggestions. We are now addressing his/her concerns one by one and meanwhile we modified the manuscript based on his/her comments.

Point 1: Section 2.4 There seems to be only one MS/MS transition per analyte, while state of the art are two transitions. DO the investigated analytes yiled only one product ion?

Response 1: Both TMA and TMAO can be monitored by two precursor to daughter transitions in positive MRM MS with different abundance, TMA: m/z 60→44, 60→45; TMAO: m/z 76→58, 76→59. In this manuscript, we used the relatively abundant MS/MS transition.

Point 2: Section 2.5 The S/N ratio has recently been discouraged for determination of LOD/LOQ by the European Reference Labs (Guidance document on the estimation of LOD and LOQ for measurements in the field of contaminants in feed and food. EUR 28099 EN. http://publications.jrc.ec.europa.eu/repository/bitstream/JRC102946/eur%2028099%20en_lod%20loq%20guidance%20document.pdf). Although not legally binding for bioanalytiucal methods, the authors should consider to use another criterion.

Response 2: We have recalculated the LOD and LOQ based on the European guidance. Since urine is water based and mainly contains some salt and small molecular metabolite waste, the matrix effect can be negligible and we calculated the LOQ from calibration standards as the lowest concentration with ≤30% deviation calculated based on calibration curve. LOD was calculated via Calibration Approach using Eq. C.